# Hyaluronic Acid-Modified and Doxorubicin-Loaded Gold Nanoparticles and Evaluation of Their Bioactivity

**DOI:** 10.3390/ph14020101

**Published:** 2021-01-28

**Authors:** Lin-Song Li, Bin Ren, Xiaojing Yang, Zhong-Chao Cai, Xue-Jie Zhao, Mei-Xia Zhao

**Affiliations:** 1Key Laboratory of Natural Medicine and Immuno-Engineering of Henan Province, Henan University, Jinming Campus, Kaifeng 475004, China; lilinsong@henu.edu.cn (L.-S.L.); 10350027@vip.henu.edu.cn (X.Y.); czc138393@sina.com (Z.-C.C.); fndxn@henu.edu.cn (X.-J.Z.); 2School of Mathematics and Statistics, Henan University, Jinming Campus, Kaifeng 475004, China; renjianliang1224@126.com

**Keywords:** hyaluronic acid, gold nanoparticles, drug carriers, theragnostic agents, imaging

## Abstract

Functionalized gold nanoparticles (AuNPs) have been successfully used in many fields as a result of having low cytotoxicity, good biocompatibility, excellent optical properties, and their ability to target cancer cells. Here, we synthesized AuNP carriers that were modified by hyaluronic acid (HA), polyethylene glycol (PEG), and adipic dihydrazide (ADH). The antitumor drug doxorubicin (Dox) was loaded into AuNP carriers and attached chemically. The Au nanocomposite AuNPs@MPA-PEG-HA-ADH-Dox was able to disperse uniformly in aqueous solution, with a diameter of 15 nm. The results of a 3-(4,5-dimethyl-2-thiazolyl)-2,5-diphenyltetrazolium bromide (MTT) assay indicated that AuNP carriers displayed very little toxicity toward cells in high doses, although the antitumor properties of Au nanocomposites were significantly enhanced. Cellular uptake experiments demonstrated that AuNPs modified with hyaluronic acid were more readily ingested by HepG2 and HCT-116 cells, as they have a large number of CD44 receptors. A series of experiments measuring apoptosis such as Rh123 and annexin V-FITC staining, and analysis of mitochondrial membrane potential (MMP) analysis, indicated that apoptosis played a role in the inhibition of cell proliferation by AuNPs@MPA-PEG-HA-ADH-Dox. Excessive production of reactive oxygen species (ROS) was the principal mechanism by which the Au nanocomposites inhibited cell proliferation, leading to apoptosis. Thus, the Au nanocomposites, which allowed cell imaging in real-time and induced apoptosis in specific cell types, represent theragnostic agents with potential for future clinical applications in bowel cancer.

## 1. Introduction

Gold nanoparticles (AuNPs) are used in cancer diagnosis and biological imaging because of their excellent optical properties, low cytotoxicity, and good biocompatibility [1,2,3,4,5]. As potential drug carriers, AuNPs have received considerable attention by many researchers [6,7,8,9,10], as it has been frequently reported that they can induce programmed cell death (i.e., apoptosis) [11,12]. Furthermore, as drug carriers, AuNPs can transport imaging and therapeutic agents, providing the possibility of diagnosis, treatment, and monitoring therapeutic progress, while AuNPs themselves do not participate in diagnosis or therapy [13,14,15,16,17]. However, AuNPs are rarely reported as agents that can inhibit cell proliferation and achieve both fluorescence-enhanced cell imaging and specific cell targeting. Therefore, improvements in functionalized AuNPs represent an important trend in cancer treatment research [18]. In the present study, we aimed to design a sensitive and specific agent using AuNPs that inhibits cell proliferation and allows fluorescence enhanced cell imaging.

Hyaluronic acid (HA) is a macromolecule from a natural source, widely used in biomedical applications due to its good biocompatibility, biodegradability, and low toxicity, especially in cosmetics, treatments for arthritis, and drug delivery [19,20,21,22,23,24]. Recent studies have demonstrated that the surface of cancer cells over-express the CD44 receptors on their surface, to which HA strongly binds [25,26,27,28,29,30]. Therefore, HA can be utilized as a target ligand for tumor-targeted drug delivery [31,32,33,34,35]. The structure of HA can be modified to produce a variety of derivatives, which not only maintains its stability, but also enhances its bio-compatibility. HA has an abundance of hydroxyl and carboxyl groups, representing multiple sites for chemical modification [36,37]. Therefore, combining AuNPs with HA represents a strategy for specifically targeting the high concentrations of CD44 on the surface of tumor cells.

Polyethylene glycol (PEG), often utilized as an adhesive, is non-toxic, highly water soluble, and is approved for use both as a food and in a variety of clinical applications by the U.S. Food and Drug Administration [38,39,40]. When combined with organic compounds, PEG can enhance the biocompatibility and increase the solubility of poorly water-soluble compounds [41,42,43]. It can inhibit the adsorption of transaminase proteins, inhibiting their clearance from blood circulation when modified with PEG, prolonging their half-life in blood, and affecting the transport of drug delivery systems [44,45,46,47]. Therefore, PEG has been widely studied in the fields of controlled drug release, wound healing, and drug delivery [48,49,50,51,52]. Doxorubicin (Dox) is an anti-tumor drug that is clinically used for the treatment of acute myelogenous leukemia, acute lymphoblastic leukemia, Hodgkin’s and non-Hodgkin’s lymphoma, breast cancer, and other malignant tumors [53,54,55,56,57,58]. However, it displays a low utilization rate, causes severe adverse reactions, and an inability to effectively reach the sites of tumors [59,60,61,62]. Although AuNPs can disperse well in aqueous solution, their duration in the circulation is not sufficiently long and they easily agglomerate [63,64,65,66]. To overcome these shortcomings, Dox, HA, and AuNPs were combined for drug delivery, having improved dispersion and enhanced absorption characteristics, allowing the transport into cells for reduced toxicity and side effects with enhanced treatment effects.

In the present study, Au nanoparticles with a diameter of 15 nm were synthesized by boiling, and their surfaces modified by adsorption, yielding Au@MPA. PEG-HA was chemically bonded then modified and linked to ADH to obtain PEG-HA-ADH, which was then chemically bonded to the AuNPs to yield Au@MPA-PEG-HA-ADH (Scheme 1). The AuNPs were loaded with the anti-tumor drug Dox and retained using hydrazone bonds to obtain the final compound Au@MPA-PEG-HA-ADH-Dox (Scheme 1), which displayed good stability in aqueous solution. The antitumor properties of the Au nanocomposite were evaluated using an MTT assay and their cellular uptake observed using a fluorescent inverted microscope. Cell apoptosis induced by the Au nanocomposites was identified by flow cytometry and high-content imaging. Cellular uptake experiments demonstrated that the Au nanocomposites were preferentially ingested by HepG2 and HCT-116 cells that expressed a large number of CD44 receptors.

## 2. Results and Discussion

### 2.1. Synthesis and Characterization

First, the ligand PEG-HA-ADH was synthesized through chemical synthesis (Scheme 2), while the ^1^Hydrogen-Nuclear Magnetic Resonance (^1^H NMR) spectrum (Appendix A) and Fourier transform infrared (FTIR) spectrum (Appendix A) were used to evaluate the reaction. From the ^1^H NMR spectrum, a peak relating to an amide bond had a chemical shift of 11, indicating that the compound PEG-HA-ADH was successfully obtained. Au particles with a diameter of 15 nm were obtained by boiling, after which PEG-HA-ADH ligands were grafted onto the surface with chemical bonds, obtaining the modified form of AuNPs, Au@MPA-PEG-HA-ADH. Furthermore, the AuNPs were loaded with the anti-tumor drug Dox and immobilized with hydrazone bonds to obtain the final compound Au@MPA-PEG-HA-ADH-Dox, which had excellent stability in aqueous solution. The AuNPs and Au nanocomposites were characterized by transmission electron microscopy (TEM), FTIR spectrum (Appendix A), ultraviolet and visible spectrophotometry (UV–Vis), and fluorescence spectroscopy. The FTIR spectrum of Dox is displayed in Appendix A. As shown in Appendix A, Au@MPA-PEG-HA-ADH-Dox had peaks of PEG, HA, ADH, and Dox. Therefore, it is probable that Dox was successfully incorporated into the MPA-HA-PEG-ADH modified Au nanoparticles. As can be observed from Figure 1, TEM indicated that Au@MPA exhibited slight aggregation behavior. After modification with HA and PEG, the AuNPs exhibited good dispersibility. After loading with Dox, the Au nanocomposites retained good dispersibility, although the particle size became quite large while a film had formed on the surface of the AuNPs, indicating that they had been modified. Dispersion of the particles was measured, as shown in Appendix A, where the variation in the dispersion of the nanoparticles after a range of resting times was displayed. Only Au@MPA displayed strong aggregation. After modification with HA and PEG, the AuNPs displayed excellent dispersibility, a characteristic that was retained after standing for 24 h.

Optical properties of the nanoparticles were characterized by UV–Vis spectroscopy. Figure 2 demonstrates that the gold nanoparticles exhibited an ultraviolet absorption peak at 520 nm, and absorption at 530 nm after modification with 3-mercaptopropionic acid (MPA). Appendix A displays the UV–Vis spectrum of Dox. The ultraviolet absorption peak of Dox shifted from 480 nm to 530 nm following its incorporation into the NPs, while the ultraviolet absorption peak of gold was red-shifted from 520 nm to 680 nm. The UV absorption peaks of all substances are shown in the final products Au@MPA-PEG-HA-ADH-Dox. Therefore, it is likely that Dox was successfully incorporated into the MPA-HA-PEG-ADH modified Au nanoparticles.

### 2.2. Cytotoxicity Testing

Cytotoxicity was measured using an MTT assay. Calculated IC_50_ values of the original and modified AuNPs are displayed in Table 1. It can be seen that the unmodified AuNPs inhibited the cell growth of normal and tumor cells to only a small extent. The ligand PEG-HA-ADH exhibited little toxicity toward cells. The IC_50_ value of Au@MPA-PEG-HA-ADH was lower than that of Au@MPA and PEG-HA-ADH because the toxicity of the nanoparticles increased with the increase in chain length. However, Dox loaded AuNPs strongly inhibited the cell growth of the tumor cells, especially AuNPs loaded with Dox, which displayed greater toxicity to HCT-116 and HepG2 cells than the other cells tested. This was because the surface of the HTC-116 and HepG2 cells had large numbers of CD44 receptors, which specifically bind HA. Conversely, Dox-loaded AuNPs inhibited cell growth to only a limited degree when cultured with normal HL-7702 cells. This indicates that AuNPs loaded with Dox displayed specific cancer cell selectivity.

### 2.3. Cellular Uptake In Vitro

Intracellular uptake of NPs was determined by confocal laser scanning microscopy. First, the intracellular uptake of Au@MPA-PEG-HA-ADH-Dox and Au@MPA-PEG-HA-Dox was compared in HepG2 cells. As shown in Figure 3, Au@MPA-PEG-HA-ADH-Dox was absorbed by the cells to a greater degree than Au@MPA-PEG-HA-Dox. This was because Dox attached to Au@MPA-PEG-HA-ADH nanoparticles via hydrazone bonds, which break easily to release Dox in an acidic environment inside specific cells. Therefore, Au@MPA-PEG-HA-ADH-Dox was selected for investigation in subsequent experiments.

The concentration-dependency of cellular uptake of Au@MPA-PEG-HA-ADH-Dox was measured. As shown in Figure 4, intracellular uptake and distribution was evaluated in Au@MPA-PEG-HA-ADH-Dox at different concentrations. The fluorescence intensity increased as the concentration of Au@MPA-PEG-HA-ADH-Dox increased. Au@MPA-PEG-HA-ADH-Dox was ingested by cells, becoming distributed within the cell cytoplasm. Subsequently, the intracellular uptake of Au@MPA-PEG-HA-ADH-Dox in different cells was assessed. Their intracellular uptake in different cells was evaluated after 6 h (Figure 5). Au@MPA-PEG-HA-ADH-Dox was ingested more by HCT-116 and HepG2 cells than other cell types, probably due to the large number of CD44 receptors on the cell membranes compared with other cells. Thus, hyaluronic acid preferentially targeted HCT-116 and HepG2 cells.

### 2.4. Apoptosis Study

It is well known that annexin V-FITC detects the early stages of apoptosis. The capability of Dox-loaded AuNPs to induce apoptosis was evaluated using annexin V-FITC staining by flow cytometry. As shown in Figure 6, the percentage of apoptotic cells in those cultured with Dox-loaded AuNPs was 47.15%, while the percentage of apoptosis induced by Dox alone was 43.06%. The percentage of apoptotic cells was significantly greater in response to the Dox-loaded AuNPs than the control cells (2.49%). Thus, Dox-loaded AuNPs were found to be more effective at inducing apoptosis.

### 2.5. Effect of Dox-Loaded Gold Nanoparticles (AuNPs) on Cell Mitochondrial Membrane Potential

Dox-loaded AuNP-mediated mitochondrial dysfunction in HepG2 cells was examined by staining the cells with JC-1 dye then analysis by flow cytometry. As shown in Figure 7, as the concentration of Dox-loaded AuNPs increased, green fluorescence moved to the right and the red fluorescence moved to the left. This indicates that Au@MPA-PEG-HA-ADH-Dox caused the mitochondrial membrane potential to decrease, while higher concentrations caused a greater decrease in membrane potential. Compared with Dox-loaded AuNPs, Dox alone did not cause a greater decrease in membrane potential. In particular, Dox caused a higher ratio of red/green fluorescence than Dox-loaded AuNPs. In other words, Dox-loaded AuNPs induced apoptosis to a greater extent than Dox alone. Compared with the control group, the red/green ratio in cells treated with Dox-loaded AuNPs increased significantly, suggesting that Dox-loaded AuNPs were capable of inducing apoptosis.

### 2.6. Change in Intracellular Reactive Oxygen Species (ROS)

The mechanism of action of 2,7-dichlorofluorescein (H_2_DCF-DA) is itself not one that is fluorescent. When entering cells, the change in reactive oxygen species (ROS) within the intracellular environment changes the color of dichlorofluorescein. When reactive oxygen species increase in concentration, H_2_DCF-DA is converted to DCF, which has a green fluorescence. The intracellular levels of ROS in HepG2 cells cultured with Au@MPA-PEG-HA-ADH-Dox at concentrations of 20, 50, and 100 μg/mL for 6 h are displayed in Figure 8. As the concentration of AuNPs increased, intracellular ROS increased correspondingly.

## 3. Materials and Methods

### 3.1. Materials

All reagents and solvents were purchased from commercial suppliers and used without further purification. HAuCl_4_ (98%) and 3-mercaptopropionic acid (MPA; 98%) were purchased from Sigma-Aldrich, while polyethylene glycol (PEG 2000) (N/A), hyaluronic acid (HA; 403, 97%), adipic dihydrazide (ADH; 98%), 1-ethyl-3-(3-dimethylaminopropyl)carbodiimide (EDC; 99%), and N-hydroxy succinimide (NHS; 98%) were purchased from J&K Scientific, while doxorubicin (Dox; 97%) was from Aladdin. Human normal hepatocytes (QSG-7701), human hepatocellular liver carcinoma cells (HepG2), cervical cancer cells (HeLa), and colon cancer cells (HCT-116) were purchased from the Chinese Academy of Sciences Shanghai Cell Bank.

### 3.2. Instruments

^1^H NMR spectra were recorded using a Bruker AV-400 spectrometer. An Arrary Scan VTI HCS 600 (Thermo, USA) was used for live-cell imaging. Analytical experiments were conducted using a multi-function microplate reader (BioTek, Winooski, VT, USA), and Fourier transform infrared (FTIR) spectra were measured with a NEXUS FTIR spectrometer (Thermo Nicolet Co., Santiago, MA, USA). Analytical instrumentation included a Varian Cary 300 BIO UV spectrophotometer, LCQDECA-XP mass spectrometer (Thermo, Santiago, MA, USA), DMi8 inverted fluorescence microscope (Leica, Weztlar, Germany), FACSCalibur flow cytometer (Becton Dickinson & Co., Franklin Lakes, NJ, USA), R1001-N low-temperature rotary evaporator (Zhengzhou Changcheng Branch Industry and Trade Co., Ltd., Zhengzhou, China), JEOL JEM-200CX TEM, and fluorescence spectrophotometer (Agilent, Palo Alto, CA, USA).

### 3.3. Synthesis of Au

A 156 μL volume of 150 mM HAuCl_4_ was dissolved in 80 mL of triple-distilled water then heated in the dark and boiled for 10 min. Sodium citrate solution (92 mg dissolved in 3 mL of triple-distilled water) was then added. The solution was stirred while boiling for 30 min, after which it was cooled to room temperature. Particles of Au were collected by centrifugation at 8000 rpm for 15 min, which were then resuspended in 2 mL triple-distilled water, then stored in a refrigerator at 4 °C.

### 3.4. Synthesis of Au@MPA

The gold nanoparticles fabricated in the previous step were uniformly suspended in 50 mL of triple-distilled water, the pH of which was adjusted to 11. Three mL of MPA solution in ethanol (0.01 M) were added and allowed to react for 2 h, which yielded Au@MPA that was collected by centrifugation at 9000 rpm for 5 min.

### 3.5. Synthesis of PEG-HA-ADH

HA (1.0 g, 2.5 mmol) was dissolved in 30 mL phosphate buffered saline (PBS) (pH = 5.0) in a 250 mL round bottomed flask, then ADH (0.1742 g, 1 mmol), EDC (0.192 g, 1 mmol), and NHS (0.119 g, 1 mmol) were added and reacted at 25 °C for 24 h within an atmosphere of nitrogen for protection. After completion of the reaction, PEG (0.040 g, 0.02 mmol), EDC (0.096 g, 0.5 mmol), and NHS (0.059 g, 0.5 mmol) were added to continue the reaction for a further 24 h within nitrogen for protection. At the end of the reaction, the products were purified using dialysis, with PEG-HA-ADH obtained by vacuum distillation using a rotary evaporator.

### 3.6. Synthesis of Au@MPA-PEG-HA-ADH

Au@MPA nanoparticles were uniformly dispersed in 50 mL triple-distilled water, the pH of which was adjusted to 6.5 using MES (2-(N-morpholino)ethanesulfonic acid, 50 mM). EDC (0.007 g) and NHS (0.011 g) were added and the reaction mixture was stirred for 30 min at 25 °C. Ten mg of PEG-HA-ADH was added and the mixture was stirred at 25 °C for 24 h. The final product, Au@MPA-PEG-HA-ADH, was collected by centrifugation at 10,000 rpm for 5 min.

### 3.7. Synthesis of Au@MPA-PEG-HA-ADH-Dox

A 6.8 mg quantity of Au@MPA-PEG-HA-ADH and 5 mg of Dox were dissolved in 15 mL absolute ethanol and reacted for 30 min. A 50 μL aliquot of trifluoroacetic acid was added after which the mixture was incubated overnight within an atmosphere of nitrogen. Au@MPA-PEG-HA-ADH-Dox was obtained by centrifugation at 12,000 rpm, after which the product was washed with absolute ethanol to remove unincorporated Dox. The final product was obtained by vacuum drying. The supernatants from all centrifugation and washing steps were collected, and the concentration of Dox was measured in order to quantify drug loading, and therefore the drug loading ratio was calculated.

The standard curve equation for Au@MPA-PEG-HA-ADH-Dox was: *y* = 39.074*x* + 2.137 (*R*^2^ = 0.9978). Three mg of Dox was added to 30 mL of the reaction solution. Final Dox concentration in the reaction solution was measured as 72.471 μg/mL, and thus final drug loading was 27.52% (Appendix A).

### 3.8. Synthesis of Au@MPA-PEG-HA-Dox

Au@MPA nanoparticles were uniformly dispersed in 50 mL of triple-distilled water, the pH of which was adjusted to 6.5 with MES (ethanesulfonic acid, 50 mM). EDC (0.007 g), and NHS (0.011 g) were added and the reaction mixture was stirred for 30 min. PEG (22 mg) was then added to the reaction mixture, which was stirred for a further 30 min. The mixture was then centrifuged to obtain Au@MPA-PEG. HA (1.0 g, 2.5 mmol) in 30 mL PBS (pH = 5.0), EDC (0.192 g, 1 mmol) and NHS (0.119 g, 1 mmol) were placed in a 250 mL round-bottomed flask, and Au@MPA-PEG added, then stirred in the dark for 24 h. Au@MPA-PEG-HA was obtained after centrifugation. Au@MPA-PEG-HA and Dox were dissolved in an appropriate volume of N,N-dimethylformamide (DMF), after which the reaction mixture was reacted within an atmosphere of N_2_ for 24 h. After completion of the reaction, centrifugation was used to yield Au@MPA-PEG-HA-Dox.

### 3.9. Antitumor Activity Assays of AuNPs

HL-7702, HeLa, HepG2, and HCT-116 cells were used to evaluate the antitumor activity of the AuNPs using a modified MTT (3-(4,5-dimethyl-2-thiazolyl)-2,5-diphenyltetrazolium bromide) assay. After incubation of the samples with cell cultures for 48 h, a Tecan Infinite F200 M200 multimode reader was used to measure the absorbance of the solutions in each well at 570 nm.

### 3.10. Cellular Uptake of AuNPs

After 6 h with different concentrations of Au@MPA-PEG-HA-ADH-Dox, HL-7702, HeLa, HepG2, and HCT-116 cells were stained with Hoechst 33342 (10 μg/mL) for 30 min. The cells were analyzed using an inverted fluorescent microscope.

### 3.11. Early Cell Apoptosis

HepG2 cells were cultured with Au@MPA-PEG-HA-Dox (50 μg/mL, 100 μg/mL) or Au@MPA-PEG-HA-ADH-Dox (50 μg/mL or 100 μg/mL) for 6 h. Cells were then stained with annexin V-FITC for 30 min. Apoptosis of the cells was measured using a FACSCalibur flow cytometer (Becton Dickinson Co.).

### 3.12. Analysis of Mitochondrial Membrane Potential (MMP)

After 24 h culture of cells with Au@MPA-PEG-HA-ADH-Dox (20 μg/mL, 50 μg/mL) at 37 °C in an incubator, HepG2 cells were then incubated with JC-1 (10 μg/mL) for 15 min. The mitochondrial membrane potential was then measured using a flow cytometer.

Twenty-four hours after culture of HepG2 cells with Au@MPA-PEG-HA-ADH-Dox (20 μg/mL, 50 μg/mL) at 37 °C in an incubator, they were then incubated with Rh123 (5 μg/mL) for 30 min prior to analysis by flow cytometry.

### 3.13. Measurement of ROS

2′,7′-dichlorodihydrofluorescein diacetate (H_2_DCF-DA) reagent was used to measure the cellular accumulation of ROS. HepG2 cells were treated with Au@MPA-PEG-HA-ADH-Dox samples (20, 50, 100 μg/mL) for 24 h, then incubated with H_2_DCF-DA (10 μM) for 30 min in a cell incubator at 37 °C. The fluorescence due to ROS was determined using a flow cytometer and fluorescence microscopy.

## 4. Conclusions

In conclusion, dispersed gold nanoparticles were synthesized by boiling. After modification with HA, PEG, and ADH, the AuNPs with a diameter of 15 nm displayed good stability in aqueous solution that better prevented agglomeration. MTT assays demonstrated that the gold nanoparticles alone were slightly toxic at high concentration. As the modified chain was increased in length, the toxicity of the nanoparticles increased. After loading with the drug Dox, the AuNPs exhibited good anti-tumor properties. Cellular uptake experiments demonstrated that AuNPs modified with HA were more readily ingested by HepG2 and HCT-116 cells, as they have a large number of CD44 receptors. A series of experiments measuring apoptosis indicated that the modified AuNP Au@MPA-PEG-HA-ADH-Dox was effective in inducing the formation of excessive intracellular ROS, causing initiation of the apoptosis process. The results demonstrate that the modified AuNPs represent a potent therapeutic agent for the treatment of different types of cancer using fluorescent imaging.

## Data Availability

Available in the main text and Appendix A.

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
