# Peer review of "Hyaluronic Acid-Modified and Doxorubicin-Loaded Gold Nanoparticles and Evaluation of Their Bioactivity"

_pharmaceuticals, 2021, doi:10.3390/ph14020101_

Round 1
Reviewer 1 Report
Dear Authors,
The manuscript presents interesting scientific research, however, it should be improved:
- Fig 2. - UV-Vis spectrum of pure Dox should be added, to see the shift of the corresponding band due to the presence of Dox in the system.
- FTIR spectra are incorrectly registered, I suggest registering the background again or subtracting it from the registered spectra. Identifying the correct bands is now difficult. The FTIR spectrum for pure DOX is missing. It should be added.
- Fig. 7 - the graphs are poorly visible. The results regarding the influence of dox on changes in the membrane potential should be described in more detail, because these unreadable graphs do not show such spectacular changes, in my opinion.
- The Materials and Methods section should be before the Results and Discussion section.
- The Conclusion should be more detailed.
Author Response
The manuscript presents interesting scientific research, however, it should be improved.
Q1: Fig 2. - UV-Vis spectrum of pure Dox should be added, to see the shift of the corresponding band due to the presence of Dox in the system.
A1: We thank the reviewer for his suggestion. We have added the UV-Vis spectrum of pure Dox in Figure S7 in Supporting Information.
Q2: FTIR spectra are incorrectly registered, I suggest registering the background again or subtracting it from the registered spectra. Identifying the correct bands is now difficult. The FTIR spectrum for pure DOX is missing. It should be added.
A2: We thank the reviewer for his suggestion. We have added the FTIR spectrum of Dox in Figure S5 in Supporting Information. And we discussed the experimental result in page 3.
Q3: Fig. 7 - the graphs are poorly visible. The results regarding the influence of dox on changes in the membrane potential should be described in more detail, because these unreadable graphs do not show such spectacular changes, in my opinion.
A3: We thank the reviewer for his suggestion. We have replaced the poorly visible picture with a clear one for Figure 7. And we have discussed the experimental result in page 6.
Q4: The Materials and Methods section should be before the Results and Discussion section.
A4: We thank the reviewer for his suggestion. We have again studied the pharmaceuticals-template carefully, and the Materials and Methods section should be after the Results and Discussion section.
Q5: The Conclusion should be more detailed.
A5: We thank the reviewer for his suggestion. We have discussed the Conclusion more detailed.

Reviewer 2 Report
The manuscript requires major revision for possible publication in Pharmaceuticals.
- The introudction should be modified to reveal the meaning of the proposed work.
- Some data appear to be intuitive but are not well-mentioned.
For example, as shown in Figure 2, there is insufficient explanation on the meaning of the peak shift
and how the peak change was successfully synthesized.
- Figure S5 shows the variation in dispersion of the nanoparticles over different resting times.
However, I think it would be better to clearly show the image of the time table before and after each processing,
rather than showing it as shown in Fig. S5.
Author Response
Q1:The introduction should be modified to reveal the meaning of the proposed work.
A1: We thank the reviewer for his suggestion. We have modified the introduction section.
Q2: Some data appear to be intuitive but are not well-mentioned. For example, as shown in Figure 2, there is insufficient explanation on the meaning of the peak shift and how the peak change was successfully synthesized.
A2: We thank the reviewer for his suggestion. We have discussed the experimental result in Figure 2.
Q3: Figure S5 shows the variation in dispersion of the nanoparticles over different resting times. However, I think it would be better to clearly show the image of the time table before and after each processing, rather than showing it as shown in Fig. S5.
A3: We thank the reviewer for his suggestion. We have observed the images of the time table before and after each processing, and the images are basically the same.

Reviewer 3 Report
In this contribution by Ren and co-workers, the authors prepared hyaluronic acid-modified and doxorubicin-loaded gold nanoparticles and investigated their bioactivity in several cancer cell lines. The results are kind of interesting and potentially attractive to the readership of Pharmaceuticals. However, lots of information are missing, and It could be publishable in due course but these points below must be addressed prior to publication.
- The IC50 of Au@MPA-PEG-HA-Dox should be added in table 1.
- The authors should add discussion about why the IC50 of Au@MPA-PEG-HA-ADH is lower than IC50 of Au@MPA and PEG-HA-ADH.
- The author should characterize how many chains of PEG and ADH on each HA by NMR. Such information needs to be included in the SI.
- The information of all chemicals should be added in details (at least purity and company) in section 3.1.
- There is problem with the unit in several places. For example, ‘50 L’ (line 244), what’s the amount of Au nanoparticle used for 1 g of HA? (line 260-262).
- Is there any DLS measurement to show the size and stability of Au@MPA-HA-PEG-ADH-Dox?
- Some recent studies (Acta biomaterialia 2019, 83, 314-321; Cells 2020, 9 (7), 1606) related to the applications of HA should be included (line 45).
- The full name of EDC is not 1,2-dichloroethane (line 226).
- The image resolution of figure 6 and 7 are not enough.
- Format issue. For example, ‘IC50’ (line 120), ‘HAuCl4’ (line 213), ‘N2’ (line 265).
- Language needs to be revised thoroughly. For example: line 54-60, line 87-89, line 120.
Author Response
Q1: The IC50 of Au@MPA-PEG-HA-Dox should be added in table 1.
A1: We thank the reviewer for his suggestion. We have added the IC50 of Au@MPA-PEG-HA-Dox in table 1.
Q2: The authors should add discussion about why the IC50 of Au@MPA-PEG-HA-ADH is lower than IC50 of Au@MPA and PEG-HA-ADH.
A2: We thank the reviewer for his suggestion. We have discussed the reason about the IC50 of Au@MPA-PEG-HA-ADH is lower than IC50 of Au@MPA and PEG-HA-ADH in page 4.
Q3: The author should characterize how many chains of PEG and ADH on each HA by NMR. Such information needs to be included in the SI.
A3: We thank the reviewer for his suggestion. Because PEG and HA are polymer, and a lot of literature has reported that when the presence of a characteristic of these substance indicates a successfully connection.
Q4: The information of all chemicals should be added in details (at least purity and company) in section 3.1.
A4: We thank the reviewer for his suggestion. We have added all chemicals in details in section 3.1.
Q5: There is problem with the unit in several places. For example, ‘50 L’ (line 244), what’s the amount of Au nanoparticle used for 1 g of HA? (line 260-262).
A5: We thank the reviewer for his suggestion. We have corrected the 50 L to 50 mL, and 1 g of HA is 0.25 mmol.
Q6: Is there any DLS measurement to show the size and stability of Au@MPA-HA-PEG-ADH-Dox?
A6: We thank the reviewer for his suggestion. Dispersion and stability of Au@MPA-HA-PEG-ADH-Dox was measured by TEM and picture, and the results showed the AuNPs displayed excellent dispersibility, a characteristic that was retained after standing for 24 h. So we didn’t measure the DLS.
Q7: Some recent studies (Acta biomaterialia 2019, 83, 314-321; Cells 2020, 9 (7), 1606) related to the applications of HA should be included (line 45).
A7: We thank the reviewer for his suggestion. We have cited these works in References 23 and 24.
Q8: The full name of EDC is not 1,2-dichloroethane (line 226).
A8: We thank the reviewer for his suggestion. We have corrected the full name of EDC to 1-Ethyl-3-(3-dimethylaminopropyl)carbodiimide.
Q9: The image resolution of figure 6 and 7 are not enough.
A9: We thank the reviewer for his suggestion. We have replaced the poorly visible picture with a clear one for Figure 6 and Figure 7.
Q10: Format issue. For example, ‘IC50’ (line 120), ‘HAuCl4’ (line 213), ‘N2’ (line 265).
A10: We thank the reviewer for his suggestion. Our WHOLE manuscript has been checked carefully and all format issues have been corrected.
Q11: Language needs to be revised thoroughly. For example: line 54-60, line 87-89, line 120.
A11: We thank the reviewer for his suggestion. Our WHOLE manuscript has been checked carefully and thoroughly refined the language by Labedit Company. The constructed sentences, grammatical errors, and spelling mistakes in the manuscript have been corrected.

Round 2
Reviewer 1 Report
I recommend this manuscript for publication in present form.
Reviewer 2 Report
I have checked the revised manuscript. The authors have addressed my comments. so I recommend this work should be published in this journal.
Reviewer 3 Report
The authors have solved all my questions in the revision. Therefore, I recommend it for publication.